# Nuclear analytical techniques used to study the trace element content of *Centaurium erythraea* Rafn, a medicinal plant species from sites with different pollution loads in Lower Silesia (SW Poland)

Anna Brudzińska-Kosior[1], Grzegorz Kosior[2]*, Monika Sporek[2], Zbigniew Ziembik[2], Inga Zinicovscaia[3,4], Marina Frontasyeva[3], Agnieszka Dołhańczuk-Śródka[2]

1 Department of Ecology, Biogeochemistry and Environmental Protection, Wrocław University, Wrocław, Poland, 2 Institute of Environmental Engineering and Biotechnology, University of Opole, Opole, Poland, 3 Division of Nuclear Physics, Department of Neutron Activation Analysis and Applied Research, Frank Laboratory of Neutron Physics, Joint Institute for Nuclear Research, Moscow Region, Russian Federation, 4 Horia Hulubei National Institute for R&D in Physics and Nuclear Engineering, Bucharest Magurele, Romania

* grzegorz.kosior@uni.opole.pl

## Abstract

*Centaurium erythraea* (Gentianaceae) is a medicinal plant species with therapeutic potential officially listed in the pharmacopoeias of many European, Asian and American countries. It has had many uses in natural medicine since ancient times and it is collected mostly from wild populations. The aim of this study is to investigate the trace element composition of *C. erythraea* using instrumental neutron activation analysis (INAA). The results of the performed investigations show that INAA has proved to be an efficient analytical technique for the determination of trace elements in medicinal plants. The studied plant contains elements important to the human diet and metabolism that are needed for growth, development and the prevention and curing of disease. A comparison with the reference levels of elements for plants shows that the concentrations of most elements in *C. erythraea* collected from all types of sites exceed those regarded as the reference. Compared to the values of the elements in *C. erythraea* from rural areas (LP), the concentrations of most of the investigated elements in *C. erythraea* collected from the lignite basin, urban areas and in the vicinity of the A4 highway (MP) were significantly higher. The results obtained can be used for control and monitoring in the production of pharmaceuticals based on natural medical plants.

## Introduction

The use of medical herbs to relieve and cure human diseases in former times, but their applications decreased with the introduction of modern medicine. Recently, the use of medicinal plants is again increasing around the world due to their features and low incidence of side effects [1–3]. Significant interest in herbal remedies and preparations is causing increasing

**Data Availability Statement:** The data underlying the results presented in the study are available from the OSF repository (osf.io/3vdys).

**Funding:** This study was carried out as part of the Joint Institute for Nuclear Research Programme (JINR-Poland). The Programme provided funding for the analysis of samples using the INAA method on the IBR-2 reactor. Role of the funders: The Joint Institute for Nuclear Research covered costs of samples preparing and analysis on the IBR-2 reactor.

**Competing interests:** The authors have declared that no competing interests exist.

demand for raw materials from medicinal plants [2]. Despite the serious degradation of the environment resulting from the dynamic development of industry in Europe and the rest of the world, the collection of plant material from natural habitats remains the main source of plants commonly used in natural medicine [4–7]. Knowledge of the elemental content of medicinal plants is very important because some of the elements are closely related to the essential set of elements in the human body [8, 9]. On the other hand, interest in the chemical composition of medicinal herb products is growing because of ongoing developments in bio-chemical surveying and in human mineral nutrition [10]. The therapeutic properties of numerous herbal plants are well documented; however, very little information is available regarding their trace element contents. It is therefore imperative to monitor these herbal plants for the presence of necessary and toxic element levels to determine their efficacy and safety prior to their direct use in phytotherapy by consumers or in formulations in the pharmaceutical industry [11].

*Centaurium erythraea* (Gentianaceae) is a medicinal plant species with therapeutic potential [2, 4]. It is widespread in Europe and parts of western Asia and northern Africa, where the aerial parts are collected [12]. This medicinal plant species has had many uses in natural medicine since ancient times [13]. The extract from the aerial parts of the plant is used as a stomachic and bitter tonic for digestive complications and to treat febrile conditions, diabetes and heartburn [13, 14]. 'Centaurii herba' is also used in traditional natural medicine to cleanse the blood and kidneys, relieve indigestion and heal wounds and sores [15, 16]. *C. erythraea* is collected mostly from wild populations [2, 4, 17]. In Poland, about 10 tons of the raw materials of *C. erythraea* are collected each year [18]. This is a large amount because the plant is relatively small, but the number of sites where it occurs is very limited. Frequently *C. erythraea* is collected from all available sites. The natural habitats of this species may be influenced by trace element pollution from local or long-range deposition [18]. Therefore, there is a need to study the biogeochemistry of *C.erythraea* to determine doses when it is ingested as a remedy and to whether this species is an accumulator of trace elements [19].

The aim of this study is to investigate the trace element composition in *C. erythraea* using instrumental neutron activation analysis (INAA). The total concentrations of the elements Al, As, Ba, Br, Ce, Cl, Co, Cr, Cs, Fe, Hf, La, Mn, Mo, Ni, Rb, Sb, Sc, Se, Sm, Sr, Tb, Th, U, V, W and Zn were determined in raw material from regions with different pollution levels: an area of the lignite mining industry with a lignite-fired power plant, urban and rural sites. Another significant aspect of the study is to classify the results and to examine by data mining analysis whether there is a relationship between the trace elements. To achieve this, co-variabilities to describe the random or non-linear relationship between concentrations, principal component analysis and cluster analysis methods were used. The results of this study allow the dosage for medicines prepared from *C. erythraea* to be specified and the how much risk different elements of it are considered to pose to consumers.

The tested hypotheses: (1) whether *C. erythraea* from sites with different anthropogenic loading is contaminated by elevated concentrations of selected trace elements, causing a risk for medicinal consumption; (2) whether statistical techniques separating the concentration of elements in this species will allow a relationship between the trace elements studied to be distinguished.

## Materials and methods

### Sampling design

In Lower Silesia, ten sampling sites of *C. erythraea* were selected: sites 1–3 and 9 in the polluted area of the Bogatynia lignite basin; sites 6–8 near the urban areas of the city of Wrocław,

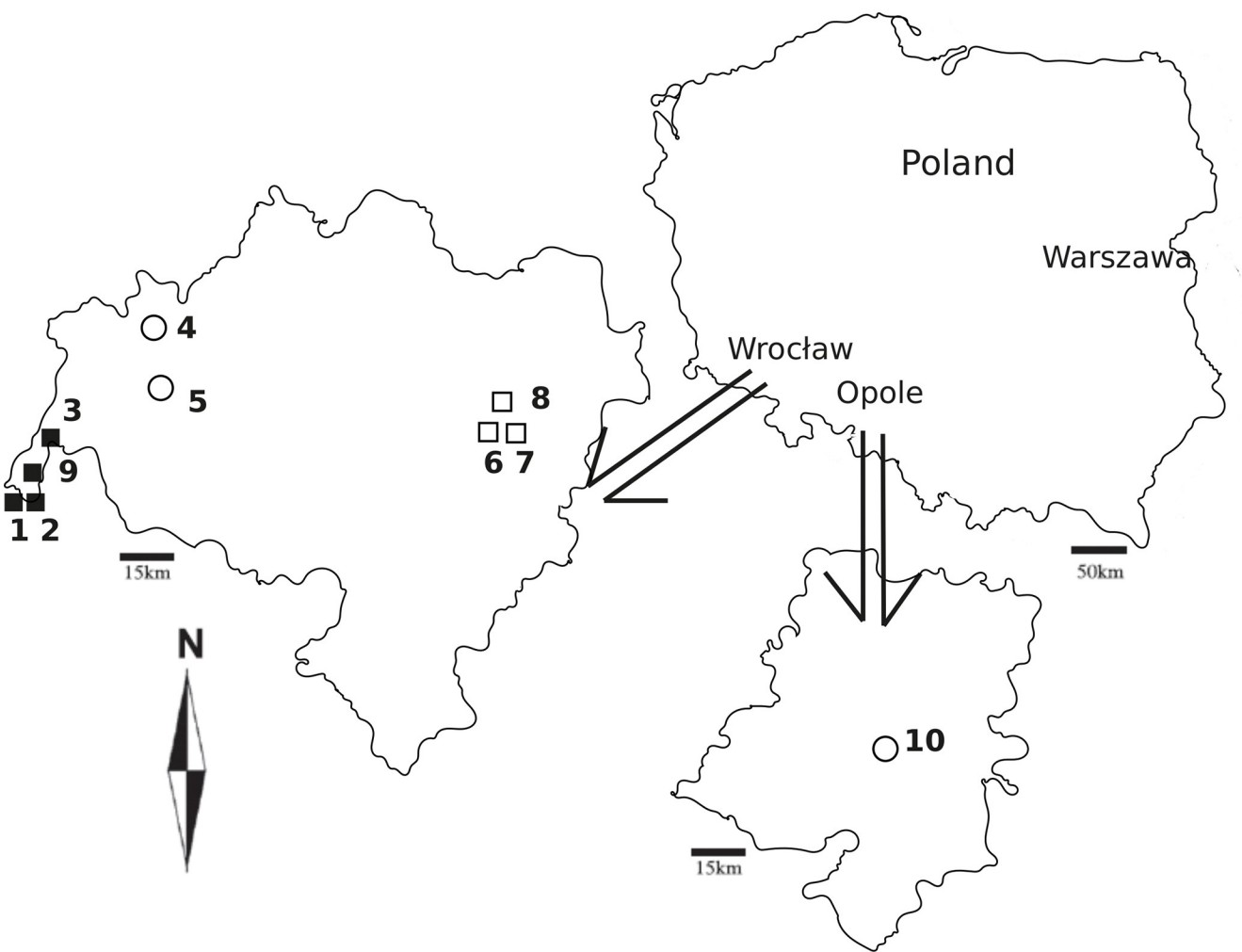

**Fig 1. Map (image by author) showing the study areas and sampling locations: ■–lignite basin sites, □–urban sites, ○–rural sites, based on [47].**

influenced mostly by urban pollutants; sites 4 and 5 in the rural areas near Bolesławiec; site 10 in the vicinity of the A4 highway, affected by transport pollutants (Fig 1). At each site, five samples of the leaves of *C. erythraea* were collected randomly within a 25 m x 5 m square. Each sample consisted of a mixture of three sub-samples. The total number of plant samples was $n$ = 10 x 5 = 50. The collected plants had not been exposed directly to canopy through fall. The centaurium leaves were wrapped in aluminium foil and transported to the laboratory.

## Sample collection and preparation

In the laboratory, the samples of *C. erythraea* were cleaned of extraneous plant materials and soil, dried at room temperature and then oven dried at 40°C for 48 hours until they were a constant weight. The samples were not washed. The dried samples were ground into fine powder in an agate mortar. Two sub-samples of each plant were pressed into thick pellets using a 15 mm diameter press-form without a binder. The plant pellets, weighing ~0.3 g, were heat-sealed in polyethylene foil bags and packed in aluminium cups for short and long irradiation, respectively [8].

## Analysis

The concentration of elements in the herbal plant samples was determined by multi-element instrumental neutron activation analysis (INAA) at the IBR-2 reactor, FLNP JINR, Dubna. The detailed procedure for sample irradiation can be found in [8, 20, 21].

Two types of irradiation were carried out. The concentrations of elements based on short-lived radionuclides: Al, Cl, Mn, and V, were determined by irradiation for 3 min and spectra measurement for 15 min. To determine the elements with long-lived isotopes–As, Br, Fe, La, Mo, Sm, U, W, Ba, Ce, Co, Cr, Cs, Hf, Ni, Rb, Sb, Sc, Se, Sr, Tb, Th, and Zn–the cadmium-screened channel 1 was used. Samples were irradiated for 4 days at a neutron flux of $1.8 \times 10^{11}$ $cm^{-2}s^{-1} \cdot min$, repacked and measured twice, first after 4 days for 30 min and then after 20 days for 1.5 hours. The gamma spectra of the induced activity were obtained using three spectrometers based on HPGe detectors with an efficiency of 40–55% and a resolution of 1.8–2.0 keV for the total-absorption peak at 1332 keV for the isotope $^{60}Co$.

The analysis of the spectra was performed using Genie2000 software from Canberra (Mirion Technologies (Canberra), Inc.), with the verification of the peak fit in an interactive mode. The concentration calculation was carried out using the 'Concentration' software developed for internal use in FLNP (Frank Laboratory of Neutron Physics) [22].

## Quality control

The quality control of the analytical measurements was carried out using certified reference materials: NIST SRM 1573 –tomato leaves, NIST SRM 1575a –pine needles, NIST SRM 1547 – peach leaves, NIST SRM 1633b –coal fly ash, IRMM BCR-667 –estuarine sediment and NIST SRM 2710 –Montana soil were irradiated under the same experimental conditions as the samples. Good agreement between the experimental results and the certified values was obtained (Table 1).

## Statistical analysis

Variance in a component concentration may not be strictly related to variance in the abundance of the component, because this variability is also affected by variabilities in the abundances of other components [23, 24]. As the result of the closure operation, the variance of a component and its covariances with the other components are bound by the relationship:

$$\mathbf{cov}(\mathbf{x_1}, \mathbf{x_2}) + \ldots + \mathbf{cov}(\mathbf{x_1}, \mathbf{x_D}) = -\mathbf{var}(\mathbf{x_1}) \tag{1}$$

Utilisation of a classical correlation estimator in the statistical analysis of compositional data, particularly, can lead to delusive conclusions [25]. An alternative method, appropriate for co-variability estimation in compositional data, is based on the variations matrix **T**. The elements $t$ of this matrix are the variances of the concentration pairs log-ratios:

$$\mathbf{t_{j1}} = \mathbf{var}\left(\mathbf{ln}\frac{\mathbf{x_j}}{\mathbf{y_1}}\right), \, \mathbf{j}, \mathbf{l} = \mathbf{1}, \ldots, \mathbf{D} \tag{2}$$

The biggest log-ratio variance is observed when concentrations follow opposite trends. Increases in one variable's values accompanied by a decrease in the values of the second one produce highly differentiated log-ratios. As the result, the biggest $t$ value is observed. To distinguish between these two types of co-variability, the terms 'positive' (an increase in both concentrations) and 'negative' (an increase in one concentration and a decrease in the second) are used.

The distribution function of $t$ can be estimated based on the actual data. Data permutations in the first variable, keeping an unchanged sequence in the second one, produce several $t$

**Table 1. Certified and experimental values for the reference materials used.**

| Element | Experimental value, <g/g | | | Certified value, <g/g | | | Z-score |
|---|---|---|---|---|---|---|---|
| Al | 252 | ± | 12 | 249 | ± | 7 | 0.19 |
| Cl | 347 | ± | 30 | 360 | ± | 19 | -0.35 |
| Sc | 37 | ± | 0.7 | 37.6 | ± | 0.6 | 0 |
| Mn | 260 | ± | 8 | 246 | ± | 7 | 1.32 |
| V | 0.68 | ± | 0.08 | 0.83 | ± | 0.008 | -1 |
| Cr | 198.2 | ± | 4.7 | 198 | ± | 4.9 | 0 |
| Fe | 100126 | ± | 4900 | 104900 | ± | 3880 | -0.76 |
| Co | 40.5 | ± | 2.4 | 42.9 | ± | 3.4 | -0.57 |
| Ni | 128 | ± | 10 | 128 | ± | 9 | 0 |
| Zn | 241 | ± | 8 | 235 | ± | 14 | 0.35 |
| As | 696 | ± | 14 | 626 | ± | 35 | 1.7 |
| Br | 99.7 | ± | 3.1 | 99.7 | ± | 2.5 | 0 |
| Rb | 124 | ± | 19 | 120 | ± | 36 | 0.09 |
| Se | 1.59 | ± | 0.15 | 1.59 | ± | 0.08 | 0.03 |
| Sr | 901 | ± | 80 | 901 | ± | 56 | 0 |
| Mo | 19 | ± | 5.7 | 19 | ± | 5.7 | 0 |
| Sb | 34.8 | ± | 1.7 | 38.4 | ± | 3 | -1.03 |
| Cs | 9.39 | ± | 0.3 | 9.39 | ± | 0.2 | 0 |
| Ba | 707 | ± | 51 | 729 | ± | 58 | 0.7 |
| La | 29.6 | ± | 1.1 | 27.8 | ± | 0.9 | 0 |
| Ce | 56.7 | ± | 2.5 | 56.7 | ± | 5.0 | 0 |
| Sm | 19.4 | ± | 3 | 19 | ± | 5.7 | 0.06 |
| Hf | 3.2 | ± | 0.9 | 3.4 | ± | 0.3 | 0.02 |
| Tb | 0.682 | ± | 0.024 | 0.682 | ± | 0.017 | 0 |
| W | 89 | ± | 15 | 93 | ± | 20 | 0.04 |
| Th | 22.6 | ± | 1.2 | 23 | ± | 4 | -0.09 |
| U | 2.26 | ± | 0.15 | 2.3 | ± | 0.12 | 0.003 |

values. Most of these correspond to a random arrangement of variable pairs. A fraction of the lowest $t$ values corresponds to concentrations positively related with each other. A fraction of the highest $t$ values corresponds to negative co-variability between concentrations [26]. In this study, the $t$ distribution quantile $q_{0.01}$ was the upper limit for the positive co-variability, and $q_{0.99}$ was the lower limit for the negative one. Though, as mentioned earlier, the Pearson correlation coefficient $r$ is generally not appropriate for co-variability assessment among concentrations, its value was calculated to compare conclusions with those related to the $t$ estimates.

The significance of differences in trace element concentrations in *C. erythraea* were calculated using R software. Levene's test of homogeneity of variance was used to verify whether the variance in the tested samples was equal.

In this present study the daily intake of metal (DIM) we determined by using equation:

$$\text{DIM} = C_{\text{trace element}} \text{ X } C_{\text{factor}} \text{ x } D_{\text{food intake}} / B_{\text{average weight}} \tag{3}$$

where

$$C_{\text{trace element}}, \ C_{\text{factor}}, \ D_{\text{food intake}}, \ B_{\text{average weight}}$$

represent the trace element concentrations in *C. erythraea* (mg/kg), conversion factor, daily intake of plant material and average body weight of humans, respectively. A conversion factor

of 0.085 was used. The average daily plant intake for adults was considered to be 0.345 kg/person/day, while the average adult body weight was considered to be 70 kg [27].

## Health risk index

An assessment of the health risk from polluted plants was calculated for Cr, Zn, Ni, Cd and Cu [28]. HRI is the ratio between exposure and the reference oral dose (RfD). An estimate of the potential hazard of trace elements to human health (HRI) through plant intake is calculated by equation:

$$HRI = DIM/RfD \tag{4}$$

where (DIM) is the daily intake of vegetable (Kg/day), RfD denotes the oral reference dose for the element (mg kg$^{-1}$ of body weight/day) [29]. The RfD values for Cr, Zn, Ni, Cd and Cu were 1.5, 0.3, 0.02, 0.001, 0.04 mg/kg/day respectively [28].

In the computations the R language [30] and functions from the libraries' 'cluster' [31] and 'compositions' [32] were used. The concentrations of elements in *C. erythraea* from more polluted (MP–lignite basin, urban areas and in the vicinity of the A4 highway) and less polluted (LP–rural areas) areas were examined with the *t*-test applied to the log-transformed data.

## Results and discussion

The concentrations of trace elements determined in *C. erythraea* are given in Table 2. The statistical parameters of the data were calculated. In Table 2, the following parameters are shown: min–minimal value, $q_1$ –lower quartile, median, arithmetic mean, $q_3$ –upper quartile, max–maximal value, SD–standard deviation, and MADN–normalised median absolute deviation about the median [33]. The MADN parameter can be regarded as a robust alternative to SD. If no outliers are in the vector of normally distributed data, then the MADN and SD values are similar to each other. Comparison of the median and mean values of concentration indicates skewness in data distribution. For Cl, Fe, Rb and Zn, the mean to median ratio exceeded 1.5. These results suppose outlying observations in the data. Compared to the average concentrations given by Markert et al. [34] for terrestrial plants (limits in mg kg$^{-1}$ in parentheses), the upper concentrations were within these ranges for the following elements: Ba, Cs, Mn, Ni, Rb, Sb, Sr and Tb in *C. erythraea* collected from all sites (Table 2). The highest concentrations of elements in the plants were determined for *C. erythraea* collected from sites affected by the lignite industry and from urban areas except for Br, Cl and Mo (the highest concentrations of these were in plants from rural areas). To assess the linear co-variability between the concentration of pairs of elements, quantiles of the *t* parameter distribution were calculated. In Table 3, the results of the computations are shown. The *r* parameters were tested for 0 value in population, on a 0.02 *p*-level. For 2D normally distributed data composed of variable pairs representing 30 cases, the critical, absolute $|r_c|$ value is 0.423. In Table 3, positive correlations bigger than *r* are marked with '+' and negative correlations smaller than *r* are marked with '-'. Positive co-variability was obtained for pairs of elements originating from anthropogenic sources and from the substrate. The elements that are regarded as microelements in the human diet (Co, Cr, Fe, Mn, Se, V and Zn) were found to have positive co-variability for multiple elements (Table 3). Negative co-variabilities were obtained for only four pairs of elements (including two with Cl).

In recent years, there has been increasing interest in the determination of the content of rare earth elements (REEs) in medicinal plants, as are these able to affect the growth and/or development of plants directly or indirectly [35]. INAA possesses an unbeatable position for the determination of REEs. In this study, the concentrations of the following REEs were

**Table 2. Statistical parameters of the element concentrations (mg kg$^{-1}$) in *C. erythraea* samples.**

| element | min | $q_1$ | median | mean | $q_3$ | max | SD | MADN | Markert* | RDA** | Microelements–human diet | Rare earth elements | element |
|---|---|---|---|---|---|---|---|---|---|---|---|---|---|
| Al | 201 | 616.8 | 777.5 | 753 | 884.5 | 1310 | 289.6 | 171.9 | 80 | | | | Al |
| As | 0.17 | 0.22 | 0.67 | 0.91 | 1.17 | 2.78 | 0.89 | 0.7 | 0.1 | | | | As |
| Ba | 14.8 | 21.6 | 29.9 | 29.3 | 32.9 | 45 | 9.9 | 10.5 | 40 | | | | Ba |
| Br | 2 | 2.7 | 3.5 | 4 | 4.2 | 10.2 | 2.4 | 1.3 | 4 | 1.5–2.5 | | | Br |
| Ce | 0.3 | 0.53 | 0.8 | 0.92 | 1.13 | 2.3 | 0.58 | 0.44 | 0.5 | | | X | Ce |
| Cl | 1470 | 1650 | 2440.5 | 2656.1 | 3452.5 | 4550 | 1107.9 | 1237.9 | 200 | | | | Cl |
| Co | 0.08 | 0.11 | 0.19 | 0.27 | 0.28 | 1.08 | 0.29 | 0.15 | 0.2 | 0.008 | X | | Co |
| Cr | 0.79 | 0.94 | 1.56 | 1.6 | 2.18 | 2.31 | 0.63 | 0.98 | 1.5 | 0.02–0.2 | X | | Cr |
| Cs | 0.05 | 0.07 | 0.09 | 0.12 | 0.14 | 0.23 | 0.07 | 0.05 | 0.2 | | | | Cs |
| Fe | 76.4 | 173 | 202.5 | 241.5 | 278 | 492 | 146.4 | 90.4 | 150 | 10–18 | X | | Fe |
| Hf | 0.01 | 0.04 | 0.08 | 0.07 | 0.1 | 0.16 | 0.05 | 0.05 | 0.05 | 18 | | | Hf |
| La | 0.1 | 0.15 | 0.4 | 0.34 | 0.48 | 0.6 | 0.18 | 0.15 | 0.2 | | | X | La |
| Mn | 33 | 47.3 | 135.5 | 102.3 | 140.3 | 152 | 51.1 | 20.8 | 200 | 1–5 | X | | Mn |
| Mo | 0.1 | 0.7 | 1.7 | 1.62 | 2.38 | 3.2 | 1.15 | 1.33 | 0.5 | | X | | Mo |
| Ni | 0.45 | 0.78 | 1.06 | 1.38 | 1.94 | 3.2 | 0.89 | 0.82 | 1.5 | 0.13–0.4 | X | | Ni |
| Rb | 3 | 4.25 | 19 | 23.4 | 26 | 82 | 24.74 | 21.5 | 50 | | | | Rb |
| Sb | 0.01 | 0.06 | 0.07 | 0.07 | 0.07 | 0.2 | 0.05 | 0.01 | 0.1 | | | | Sb |
| Sc | 0.01 | 0.06 | 0.09 | 0.09 | 0.13 | 0.16 | 0.05 | 0.06 | 0.02 | | | X | Sc |
| Se | 0.11 | 0.12 | 0.17 | 0.3 | 0.43 | 0.85 | 0.24 | 0.09 | 0.02 | 0.02–0.2 | X | | Se |
| Sm | 0.02 | 0.03 | 0.07 | 0.06 | 0.1 | 0.11 | 0.04 | 0.04 | 0.04 | | | X | Sm |
| Sr | 10.1 | 12.9 | 16 | 16.6 | 20.9 | 24.7 | 5 | 6.5 | 50 | | | | Sr |
| Tb | 0.003 | 0.004 | 0.005 | 0.007 | 0.01 | 0.014 | 0.004 | 0.003 | 0.008 | | | X | Tb |
| Th | 0.01 | 0.04 | 0.08 | 0.1 | 0.17 | 0.23 | 0.08 | 0.1 | 0.005 | | | | Th |
| U | 0.01 | 0.025 | 0.045 | 0.048 | 0.068 | 0.11 | 0.031 | 0.037 | 0.01 | | | | U |
| V | 0.14 | 0.61 | 0.82 | 0.73 | 0.9 | 1.2 | 0.33 | 0.29 | 0.5 | 0.1–0.4 | X | | V |
| W | 0.1 | 0.15 | 0.4 | 0.36 | 0.4 | 0.8 | 0.23 | 0.22 | 0.2 | | | | W |
| Zn | 23 | 37.3 | 52.5 | 61.1 | 64.5 | 130 | 38 | 25.2 | 50 | 15 | X | | Zn |

* [34]

** RDA–Recommended Daily Dietary Allowance (in mg) [41]

determined: Ce, La, Sc, Sm and Tb. The average concentration of these elements for all sites is higher than the reference plant values [34]. However, it must be noted that the average concentrations of REEs are influenced, to a greater extent, by those plant collection places at sites located near the lignite industry and, to a lesser extent, at sites located in urban areas (Table 3). Concentrations of most REEs (e.g. La, Ce, Pr, Nd, Sm, and Eu) can exert positive or negative physiological effects on plants depending on the dosage and other conditions [36]. However, if the concentration of REEs exceeds the optimum level, they can inhibit plant growth and even cause mortality [35]. It is believed that REEs can regulate the absorption of other mineral elements by plants. In some cases, the lowest *t* value in the complete data (positive co-variabilities, increase–increase) was found for pairs of REEs such as La–Sm, La–Tb, Sm–Tb (Table 3). Compared to the values of elements in *C. erythraea* from rural areas (LP), the concentrations of most of the investigated elements in *C. erythraea* collected from the lignite basin, urban areas and in the vicinity of the A4 highway (MP) were significantly higher (test *t*, $p < 0.05$) (Table 4). The daily intake of trace elements in *C. erythraea* for adults is given in Table 5. The daily intake of metals in the consumption of plants grown in more polluted sites (MP) was significantly higher. It is impossible to determine whether there is a risk of consuming *C.*

**Table 3. Covariabilities in concentration pairs estimated with the Pearson correlation coefficient (above the diagonal) and the t quantile encoded "r" for random or non-linear "r", "+" for positive and "-" for negative covariabilit.**

| | Al | As | Ba | Br | Cd | Ce | Cl | Co | Cr | Cs | Cu | Fe | Hf | La | Mn | Mo | Ni | Pb | Rb | Sb | Sc | Se | Sm | Sr | Tb | Th | U | V | W | Zn |
|---|---|---|---|---|---|---|---|---|---|---|---|---|---|---|---|---|---|---|---|---|---|---|---|---|---|---|---|---|---|---|
| Al | | r | r | r | | R | r | r | r | r | | r | r | r | r | r | r | | r | r | r | r | r | r | + | r | + | r | r | r |
| As | 0.89 | | r | r | | R | r | + | r | + | | r | r | r | r | r | r | | + | r | r | r | r | r | + | + | + | r | r | r |
| Ba | 0.49 | | | r | | R | r | r | r | r | | r | r | r | r | r | r | | - | r | r | r | r | r | r | + | + | r | r | r |
| Br | 0.42 | 1.74 | | | | R | r | r | r | r | | r | r | r | r | r | r | | r | r | r | r | r | r | r | r | r | r | r | r |
| Cd | | | | | | | | | | | | | | | | | | | | | | | | | | | | | | |
| Ce | 0.26 | 1.52 | 0.33 | | | | r | r | r | r | | r | r | r | r | r | r | | r | r | r | r | - | r | r | r | r | r | r | r |
| Cl | 0.67 | 0.89 | 0.68 | 0.44 | | 0.68 | | r | r | r | | r | r | r | r | r | r | | r | - | r | r | - | r | r | r | r | r | r | r |
| Co | 0.47 | 1.44 | 0.29 | 1.05 | | 0.65 | 1.2 | | + | r | | r | r | r | r | r | r | | + | r | r | + | + | r | r | + | + | r | r | r |
| Cr | 0.19 | 0.29 | 0.48 | 0.59 | | 0.27 | 0.5 | 0.24 | | + | | r | r | r | + | r | r | | r | r | r | + | r | r | r | + | + | + | r | r |
| Cs | 0.22 | 0.48 | 0.71 | 0.7 | | 0.23 | 0.73 | 0.15 | 0.08 | | | r | r | r | r | r | r | | + | + | r | + | + | r | + | + | + | r | r | r |
| Cu | | | | | | | | | | | | | | | | | | | | | | | | | | | | | | |
| Fe | 0.28 | 0.39 | 0.44 | 0.51 | | 0.6 | 0.95 | 0.64 | 0.39 | 0.42 | | | + | r | r | r | r | | + | r | r | r | r | r | r | r | r | r | r | r |
| Hf | 0.61 | 1.07 | 1.02 | 1.15 | | 0.91 | 1.49 | 0.88 | 0.55 | 0.69 | | 0.32 | | + | r | r | r | | r | + | r | r | + | r | + | r | + | r | r | r |
| La | 0.36 | 0.97 | 0.77 | 0.97 | | 0.59 | 1.08 | 0.33 | 0.24 | 0.21 | | 0.19 | 0.32 | | + | r | r | | r | + | r | r | + | r | + | + | + | + | r | r |
| Mn | 0.41 | 0.57 | 0.53 | 0.64 | | 0.78 | 0.75 | 0.41 | 0.25 | 0.38 | | 0.21 | 0.4 | 0.22 | | r | r | | r | + | r | r | r | r | r | r | r | r | r | r |
| Mo | 2.13 | 0.62 | 2.51 | 1.82 | | 1.64 | 1.41 | 2.52 | 2.02 | 1.96 | | 3.31 | 4.05 | 3.29 | 3.07 | | r | | r | r | r | r | r | r | r | r | r | - | r | r |
| Ni | 0.82 | 2.48 | 0.63 | 0.89 | | 1.07 | 0.68 | 1.54 | 0.71 | 0.99 | | 0.64 | 0.57 | 0.77 | 0.62 | 3.07 | | | r | r | r | r | r | r | r | r | r | r | r | r |
| Pb | | | | | | | | | | | | | | | | | | | | | | | | | | | | | | |
| Rb | 1.5 | 1.52 | 2.18 | 2.03 | | 1.04 | 1.61 | 0.67 | 0.82 | 0.65 | | 1.96 | 2.01 | 1.27 | 1.51 | 1.74 | 2.31 | | | r | r | r | r | r | r | r | r | r | r | r |
| Sb | 0.47 | 0.47 | 0.71 | 0.68 | | 0.69 | 1.15 | 0.7 | 0.51 | 0.56 | | 0.13 | 0.24 | 0.35 | 0.34 | 3.7 | 0.9 | | 1.98 | | r | r | + | r | + | r | r | r | r | r |
| Sc | 0.48 | 1.08 | 1.05 | 1.41 | | 0.88 | 0.98 | 0.93 | 0.6 | 0.7 | | 0.69 | 0.84 | 0.49 | 0.65 | 3.44 | 1.01 | | 2.15 | 1.11 | | r | r | r | r | r | + | + | r | r |
| Se | 0.46 | 1.14 | 0.93 | 1.17 | | 0.52 | 1.06 | 0.3 | 0.17 | 0.21 | | 0.78 | 0.76 | 0.39 | 0.64 | 2.39 | 1.11 | | 0.69 | 0.86 | 0.87 | | r | r | r | + | + | r | r | + |
| Sm | 0.25 | 0.62 | 0.63 | 0.79 | | 0.54 | 0.32 | 0.32 | 0.24 | 0.21 | | 0.11 | 0.37 | 0.05 | 0.23 | 3.2 | 0.85 | | 1.44 | 0.25 | 0.57 | 0.44 | | r | r | + | + | r | r | r |
| Sr | 0.25 | 0.74 | 0.28 | 0.29 | | 0.45 | 0.73 | 0.42 | 0.11 | 0.24 | | 0.35 | 0.68 | 0.43 | 0.2 | 1.89 | 0.59 | | 1.16 | 0.51 | 0.87 | 0.44 | 0.36 | | r | r | r | r | r | r |
| Tb | 0.2 | 0.72 | 0.49 | 0.71 | | 0.22 | 1.76 | 0.52 | 0.16 | 0.19 | | 0.22 | 0.37 | 0.18 | 0.34 | 2.6 | 0.61 | | 1.31 | 0.34 | 0.48 | 0.41 | 0.17 | 0.32 | | + | + | r | r | r |
| Th | 0.67 | 0.77 | 1.44 | 1.58 | | 0.8 | 1.27 | 0.63 | 0.23 | 0.49 | | 0.5 | 0.21 | 0.21 | 0.56 | 3.86 | 0.96 | | 1.44 | 0.49 | 0.73 | 0.63 | 0.32 | 0.87 | 0.34 | | + | r | r | r |
| U | 0.34 | 0.59 | 1.07 | 1.07 | | 0.51 | 0.88 | 0.25 | 0.26 | 0.17 | | 0.36 | 0.31 | 0.1 | 0.37 | 2.94 | 0.92 | | 0.99 | 0.41 | 0.65 | 0.29 | 0.15 | 0.47 | 0.23 | 0.12 | | r | r | r |
| V | 0.28 | 0.38 | 0.65 | 0.96 | | 0.69 | 0.59 | 0.48 | 0.55 | 0.38 | | 0.34 | 0.48 | 0.19 | 0.25 | 3.24 | 0.78 | | 1.66 | 0.61 | 0.16 | 0.44 | 0.23 | 0.43 | 0.29 | 0.52 | 0.34 | | r | r |
| W | 0.93 | 1.33 | 0.67 | 0.92 | | 0.63 | 0.59 | 1.38 | 0.55 | 0.81 | | 0.96 | 0.84 | 1 | 0.89 | 2.21 | 0.44 | | 1.47 | 0.96 | 1.54 | 0.84 | 1.05 | 0.56 | 0.54 | 1.11 | 0.98 | 1.14 | | r |
| Zn | 0.34 | 0.74 | 0.59 | 0.56 | | 0.62 | 0.75 | 0.33 | 0.18 | 0.31 | | 0.5 | 0.63 | 0.44 | 0.34 | 2.41 | 0.92 | | 1.12 | 0.52 | 1.01 | 0.26 | 0.41 | 0.19 | 0.5 | 0.85 | 0.42 | 0.42 | 0.85 | |
| | Al | As | Ba | Br | Cd | Ce | Cl | Co | Cr | Cs | Cu | Fe | Hf | La | Mn | Mo | Ni | Pb | Rb | Sb | Sc | Se | Sm | Sr | Tb | Th | U | V | W | Zn |

**Table 4. Mean values (mg kg1), SD of concentrations of elements in *C. erythraea* from more polluted (MP–lignite basin, urban areas and in the vicinity of the A4 highway) and less polluted (LP–rural areas) areas and results of the t-test comparision.**

| Element | | Less polluted (LP) | | More polluted (MP) | | t | p | |
|---|---|---|---|---|---|---|---|---|
| | | Mean | SD | Mean | SD | | | |
| Al | | 436 | 183 | 846 | 214 | -4.3 | <0.001 | |
| Cl | | 4100 | 545 | 2321 | 871 | 4.7 | <0.001 | |
| Sc | | 0.07 | 0.05 | 0.1 | 0.04 | -1.4 | 0.176 | |
| V | | 0.38 | 0.26 | 0.85 | 0.28 | -3.8 | <0.001 | |
| **Cr** | | **0.85** | **0.07** | **1.79** | **0.53** | **-4.3** | **<0.001** | * |
| **Mn** | | **45.9** | **10** | **111** | **37** | **-4.2** | **<0.001** | * |
| **Fe** | | **78.6** | **2** | **267** | **100** | **-4.6** | **<0.001** | * |
| **Ni** | | **0.49** | **0.023** | **1.6** | **0.68** | **-4** | **<0.001** | * |
| Co | | 0.088 | 0.008 | 0.33 | 0.3 | -2 | 0.06 | |
| **Zn** | | **23.5** | **1.9** | **65** | **31** | **-3.2** | **0.003** | ** |
| **As** | | **0.2** | **0.02** | **1.1** | **0.9** | **-2.4** | **0.02** | * |
| **Se** | | **0.11** | **0.007** | **0.37** | **0.26** | **-2.5** | **0.02** | * |
| **Br** | | **2.62** | **0.26** | **4.27** | **2.4** | **-1.7** | **0.11** | * |
| Rb | | 12.2 | 9.3 | 25.1 | 22.5 | -1.4 | 0.19 | |
| **Sr** | | **10.1** | **0.29** | **18.5** | **4** | **-5** | **<0.001** | * |
| Mo | | 2.03 | 0.53 | 1.7 | 1 | 0.78 | 0.4 | |
| Sb | | 0.03 | 0.01 | 0.097 | 0.039 | -4.2 | <0.001 | |
| **Ba** | | **26.1** | **4.8** | **30.1** | **10.4** | **-0.9** | **0.38** | * |
| **Cs** | | **0.06** | **0.008** | **0.14** | **0.06** | **-3.4** | **0.002** | * |
| **La** | | **0.1** | **0.002** | **0.45** | **0.16** | **-5.5** | **<0.001** | * |
| **Ce** | | **0.56** | **0.06** | **1.08** | **0.58** | **-2.2** | **0.04** | * |
| **Sm** | | **0.02** | **0.006** | **0.08** | **0.02** | **-6.2** | **<0.001** | * |
| **Tb** | | **0.003** | **0.001** | **0.009** | **0.003** | **-3.8** | **<0.001** | * |
| **Hf** | | **0.02** | **0.012** | **0.09** | **0.034** | **-5** | **<0.001** | * |
| W | | 0.29 | 0.14 | 0.42 | 0.2 | -1.5 | 0.15 | |
| **Th** | | **0.015** | **0.004** | **0.13** | **0.07** | **-4.3** | **<0.001** | * |
| **U** | | **0.017** | **0.005** | **0.07** | **0.03** | **-4.3** | **<0.001** | * |

* Levene's test is significant ($p < 0.05$), suggesting a violation of the equal variance assumption

*erythraea* due to the lack of data. HRI-based risk assessment of studied trace elements is provided in Table 6. This method does give a reliable indication of the human health risk from exposure to pollutants. When HRI < 1, there is no obvious risk from the xenobiotic over a lifetime of exposure, while HRI > 1, the toxicant may produce a serious effect [29]. The probability of experiencing long-term carcinogenic effects increases with the HRI value [29]. The HRI index showed that the consumption of the tested plant may be a threat for some of the elements included in the index (Cd, Cu and Zn) from less polluted sites and for all elements included in the index except Cr. In this study, the content of trace elements (Co, Cr, Fe, Mn, Mo, Ni, Se, V and Zn) regarded as microelements in the human diet was determined. The total amount of these elements in the plant permits us to conclude that *C. erythraea* can be considered an important their source for humans (especially Co, Fe, Se, V and Zn), even after preparing infusions using raw plant material. These elements play an important role in the mineral nutrition of humans. For instance, Fe is useful in the prevention of anaemia and other related diseases and this element is contained in the haemoglobin of human erythrocytes. It is needed

**Table 5. Daily intake of trace elements (DIM) of *C. erythraea* (mg/kg) from more polluted (MP–lignite basin, urban areas and in the vicinity of the A4 highway) and less polluted (LP–rural areas) areas.**

|  | Al | Cl | Sc | V | Cr | Mn | Fe | Ni | Co | Zn |
|---|---|---|---|---|---|---|---|---|---|---|
| LP | 180 | 172 | 0.03 | 0.2 | 0.4 | 19 | 33 | 0.2 | 0.04 | 10 |
| MP | 350 | 970 | 0.04 | 0.4 | 0.7 | 50 | 110 | 0.7 | 0.1 | 30 |
|  | Se | Br | Rb | Sr | Mo | Sb | Ba | Cs | La | Ce |
| LP | 0.05 | 1 | 5 | 4 | 0.9 | 0.01 | 10 | 0.03 | 0.04 | 0.2 |
| MP | 0.2 | 2 | 10 | 8 | 0.7 | 0.04 | 10 | 0.06 | 0.2 | 0.5 |
|  | Tb | Hf | W | Th | U | Pb | Cu | Cd | Sm | As |
| LP | 0.001 | 0.008 | 0.1 | 0.006 | 0.007 | 0.5 | 3 | 0.02 | 0.008 | 0.09 |
| MP | 0.004 | 0.004 | 0.2 | 0.05 | 0.03 | 0.8 | 5 | 0.08 | 0.03 | 0.5 |

for a healthy immune system and for energy production [37]. Co, also essential for human health, is a key constituent of cobalamin, also known as vitamin B12, and deficiency results when fewer red blood cells, which are necessary for the rapid transport of oxygen to injured cells (vitamin B12 assists in repairing damaged axons and regenerating nerve cells), are produced in the body [38]. Although the biological role of V is limited and the available reported literature is mostly about the beneficial role in the management of diabetes mellitus, the function of V is probably a part of the system protecting against injury to tissues [37]. Zinc is useful for protein synthesis, normal body development and recovery from illness [39]. Particular attention should be paid to selenium, because it is an important element in the treatment of human tumours. Depending on the concentration and chemical form, Se may prevent or treat tumours. The inhibitory effects are tumour-specific and Se induces apoptosis in malignant cells at concentrations that do not affect the viability of normal cells [40]. Se levels in *C. erythraea* from all the studied sites (0.3 mg kg$^{-1}$) were significantly higher than the average concentrations given by [34] for terrestrial plants; therefore *C. erythraea* can be considered relatively rich in Se. However, the average concentration in plants collected from lignite industry sites was 0.5 mg kg$^{-1}$. A daily intake of *C. erythraea* can cause an overdose of Se (the recommended daily dietary allowance is 0.02–0.2 mg kg$^{-1}$) [41]. Elements e.g. Zn, Fe, Cu, Cr, Mo, Se and Co, are considered essential in human nutrition and become harmful only at high concentrations [42, 43]. With the tendency of people to consume health-promoting foods based on their distrust of medicines, preparations of trace elements are available commercially, and inappropriate use of these preparations can cause excessive states of these trace elements [44]. Persons more susceptible to the harmful effects of the uptake of low doses of heavy metals through their diet, like children, pregnant or elderly people, might show adverse reactions [3]. It should be emphasised that the contaminated sites achieved a higher concentration of these elements, but these are not dangerous levels except for the lignite sites. Unfortunately, the maximum allowable levels of elements in medicinal plant materials and the final herbal products are not regulated globally. The World Health Organization (WHO) mentions the maximum permissible levels in raw plant materials only for Cd and Pb, namely 0.3 mg kg$^{-1}$ and 10 mg kg$^{-1}$, respectively [45]. The NSF (National Sanitation Foundation) gives the following

**Table 6. Health Risk Index of trace elements (HRI) of *C. erythraea* (mg/kg) from more polluted (MP–lignite basin, urban areas and in the vicinity of the A4 highway) and less polluted (LP–rural areas) areas.**

|  | Cr | Ni | Zn | Cu | Cd |
|---|---|---|---|---|---|
| LP | 0.02 | 1 | 3 | 8 | 2 |
| MP | 0.05 | 3 | 9 | 13 | 8 |

values for As– 5 mg kg$^{-1}$, Cd– 0.3 mg kg$^{-1}$, Cr– 2 mg kg$^{-1}$ and Pb– 10 mg kg$^{-1}$, (NSF, 2001). In our study we investigated only As and Cr, and the concentration of these elements in plants collected from lignite industry sites was 1.7 mg kg$^{-1}$ for As and 1.8 mg kg$^{-1}$ for Cr, and did not exceed the permissible level.

Another reason for undertaking precise and accurate elemental analysis of medicinal plant formulations is the need to assess and monitor the safety and quality of these products (Pohl et al., 2016), primarily in reference to the content of elements that may cause negative effects on the human body, that is, Al, As, Ba, Cd, Ni, Pb, Sb and Sn: potentially exhibiting toxic effects. This can lead to avoiding over-consumption and intoxication [46]. In this study, the concentrations for Al, As, Ba, Cd, Ni, Pb, Sb and Sn elements were higher than the average for plants except for Ba and Sb [34].

## Conclusions

The results of the investigations performed show that INAA has proved to be an efficient analytical technique for determining the trace elements in medicinal plants. The studied plant–*C. erythraea*–contains elements of importance in the human diet and metabolism that are needed for growth, development and the prevention and curing of disease. The levels of trace elements in plants are affected by the geochemical characteristics of the soil and the ability of plants to selectively accumulate some of the elements. Comparison with the 'reference plant' shows that the concentrations of most elements in *C. erythraea* collected from all types of sites exceed those regarded as the reference. Compared to the values of elements in *C. erythraea* from rural areas (LP), the concentrations of most of the investigated elements in *C. erythraea* collected from the lignite basin, urban areas and in the vicinity of the A4 highway (MP) were significantly higher. The results obtained can be used for control and monitoring in the production of pharmaceuticals based on natural medical plants.

## Author Contributions

**Conceptualization:** Anna Brudzińska-Kosior.

**Data curation:** Grzegorz Kosior, Zbigniew Ziembik.

**Formal analysis:** Inga Zinicovscaia, Marina Frontasyeva.

**Methodology:** Grzegorz Kosior, Zbigniew Ziembik, Inga Zinicovscaia, Marina Frontasyeva.

**Project administration:** Grzegorz Kosior.

**Visualization:** Monika Sporek.

**Writing – original draft:** Grzegorz Kosior.

**Writing – review & editing:** Anna Brudzińska-Kosior, Monika Sporek, Agnieszka Dołhańczuk-Śródka.

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
