## [Decision Letter · Decision Letter 0]

11 Jan 2022

PONE-D-21-34064Nuclear analytical techniques used to study trace elements content of Centaurium erythraea Rafn. medicinal plant from sites with different pollution load in the Lower Silesia (SW Poland).PLOS ONE

Dear Dr. Kosior,

Thank you for submitting your manuscript to PLOS ONE. After careful consideration, we feel that it has merit but does not fully meet PLOS ONE’s publication criteria as it currently stands. Therefore, we invite you to submit a revised version of the manuscript that addresses the points raised during the review process.

The submitted file does not contain tables. Since they were submitted as "Other", they have never reached reviewers. In the revised file, please embed tables into the main text or submit them as "Supporting material". The language usage can be improved. I strongly encourage the authors to go throughout the text once again, preferably along with a native English speaker, to correct syntax, interpunction, and typos. Also note that the species authority is written Rafn (without full stop) as it refers to Carl Gottlob Rafn, a Danish botanist. References regarding biological activities of *C. erythraea* are quite outdated. Please check for updates.

We look forward to receiving your revised manuscript.

Kind regards,

Branislav T. Šiler, Ph.D.

Academic Editor

PLOS ONE

Journal Requirements:

“This study was carried out in the frame of the Joint Institute for Nuclear Research Programme”

Please note that funding information should not appear in other areas of your manuscript. We will only publish funding information present in the Funding Statement section of the online submission form.

“This study was carried out in the frame of the Joint Institute for Nuclear Research Programme (JINR-Poland).”

“This study was carried out in the frame of the Joint Institute for Nuclear Research Programme (JINR-Poland).”

4. Please amend the manuscript submission data (via Edit Submission) to include authors “Grzegorz Kosiora, Monika Sporeka, Zbigniew Ziembika, Inga Zinicovscaiabc, Marina Frontasyevab, Agnieszka Dołhańczuk-Śródkaa”.

5. We note that Figures 1 and 2 in your submission contain [map/satellite] images which may be copyrighted. All PLOS content is published under the Creative Commons Attribution License (CC BY 4.0), which means that the manuscript, images, and Supporting Information files will be freely available online, and any third party is permitted to access, download, copy, distribute, and use these materials in any way, even commercially, with proper attribution. For these reasons, we cannot publish previously copyrighted maps or satellite images created using proprietary data, such as Google software (Google Maps, Street View, and Earth). For more information, see our copyright guidelines: http://journals.plos.org/plosone/s/licenses-and-copyright.

   a. You may seek permission from the original copyright holder of Figures 1 and 2 to publish the content specifically under the CC BY 4.0 license. 

Additional Editor Comments:

The submitted file does not contain tables. Since they were submitted as "Other", they have never reached reviewers. In the revised file, please embed tables into the main text or submit them as "Supporting material". The language usage can be improved. I strongly encourage the authors to go throughout the text once again, preferably along with a native English speaker, to correct syntax, interpunction, and typos. Also note that the species authority is written Rafn (without full stop) as it refers to Carl Gottlob Rafn, a Danish botanist. References regarding biological activities of *C. erythraea* are quite outdated. Please check for updates.

Reviewers' comments:

Reviewer's Responses to Questions

**Comments to the Author**

1. Is the manuscript technically sound, and do the data support the conclusions?

Reviewer #1: Yes

Reviewer #2: No

2. Has the statistical analysis been performed appropriately and rigorously? 

Reviewer #1: Yes

Reviewer #2: N/A

3. Have the authors made all data underlying the findings in their manuscript fully available?

Reviewer #1: Yes

Reviewer #2: No

4. Is the manuscript presented in an intelligible fashion and written in standard English?

Reviewer #1: Yes

Reviewer #2: Yes

5. Review Comments to the Author

Reviewer #1: The results of this research described that investigation of the trace elements composition by the method of instrumental neutron activation analysis. Medicinal plant Centaurium erythraea was used as model plant in this experimental setup. The authors showed that concentrations of the investigated elements were significantly higher in centaury plants collected from lignite basin, urban areas and vicinity of highway in comparison to rural areas. Considering obtained results, it can be concluded that this efficient analytical technique can be used in determination of trace elements in plants. In general, the manuscript is very well written and interesting. This manuscript could have significant contribution in control and monitoring of medicinal plants production.

Minor remarks are marked in Track Changes in original text manuscript.

I have also checked the reference through the text manuscript as well as the reference list and I found a lot of mistakes. Some references are cited in the text manuscript but not in the reference list and vice versa. Also, the majority of the references are not cited properly. The authors need to correct this.

Reviewer #2: The authors submitted the manuscript with a study of trace elements content of C. erythtaea plant using INAA. The proposed manuscript does not contain the tables. The authors must include tables 1-4 mentioned in the manuscript.

6. PLOS authors have the option to publish the peer review history of their article (what does this mean?). If published, this will include your full peer review and any attached files.

Reviewer #1: No

Reviewer #2: No

---

## [Author Response · Author response to Decision Letter 0]

13 Mar 2022

Dear Editor,

Please find corrected enclosed an original manuscript entitled “Nuclear analytical techniques used to study the trace element content of Centaurium erythraea Rafn, a medicinal plant from sites with different pollution loads in Lower Silesia (SW Poland)” by Anna Brudzińska-Kosior, Grzegorz Kosior, Monika Sporek, Zbigniew Ziembik, Inga Zinicovscaia, Marina Frontasyeva, Agnieszka Dołhańczuk-Śródka.

Manuscript was corrected according to reviewers suggestions:

Reviewer: The submitted file does not contain tables. Since they were submitted as "Other", they have never reached reviewers. In the revised file, please embed tables into the main text or submit them as "Supporting material". done

Reviewer: The language usage can be improved. I strongly encourage the authors to go throughout the text once again, preferably along with a native English speaker, to correct syntax, interpunction, and typos. The text has been checked for language by a native speaker

Reviewer: Also note that the species authority is written Rafn (without full stop) as it refers to Carl Gottlob Rafn, a Danish botanist. done

Reviewer: References regarding biological activities of C. erythraea are quite outdated. Please check for updates.

Unfortunately, the literature on the species studied is extremely scarce, therefore we used the current literature on medicinal plants.

Your's sincerely 

Grzegorz Kosior

Institute of Environmental Engineering and Biotechnology

Opole University

ul. Kard. Kominka 6a, 45-035 Opole

tel: +48/781797137

---

## [Decision Letter · Decision Letter 1]

24 Aug 2022

PONE-D-21-34064R1Nuclear analytical techniques used to study the trace element content of Centaurium erythraea Rafn, a medicinal plant from sites with different pollution loads in Lower Silesia (SW Poland)PLOS ONE

Dear Dr. Kosior,

Thank you for submitting your manuscript to PLOS ONE. After careful consideration, we feel that it has merit but does not fully meet PLOS ONE’s publication criteria as it currently stands. Therefore, we invite you to submit a revised version of the manuscript that addresses the points raised during the review process.

The authors should clarify some vague sentences in Abstract and Introduction. References should be updated as suggested by Reviewers #3 and #4. Additional calculations are needed, as suggested by Reviewer #3.

I'm quite convinced that the authors have overseen my recommendations provided in the previous review round. They state: "The language usage can be improved. I strongly encourage the authors to go throughout the text once again, preferably along with a native English speaker, to correct syntax, interpunction, and typos (e.g., the statement "This species is a plant..." (L59) cannot stand from the scientific point of view; there are many similar examples). References regarding biological activities of *C. erythraea* are quite outdated. Please check for updates." Indeed, the "freshest" reference dealing with this topic dates from 2015 and to date there have been numerous articles published.

We look forward to receiving your revised manuscript.

Kind regards,

Branislav T. Šiler, Ph.D.

Academic Editor

PLOS ONE

Journal Requirements:

Additional Editor Comments (if provided):

The authors should clarify some vague sentences in Abstract and Introduction. References should be updated as suggested by Reviewers #3 and #4. Additional calculations are needed, as suggested by Reviewer #3.

I'm quite convinced that the authors have overseen my recommendations provided in the previous review round. They state: "The language usage can be improved. I strongly encourage the authors to go throughout the text once again, preferably along with a native English speaker, to correct syntax, interpunction, and typos (e.g., the statement "This species is a plant..." (L59) cannot stand from the scientific point of view; there are many similar examples). References regarding biological activities of *C. erythraea* are quite outdated. Please check for updates." Indeed, the "freshest" reference dealing with this topic dates from 2015 and to date there have been numerous articles published.

Reviewers' comments:

Reviewer's Responses to Questions

**Comments to the Author**

1. If the authors have adequately addressed your comments raised in a previous round of review and you feel that this manuscript is now acceptable for publication, you may indicate that here to bypass the “Comments to the Author” section, enter your conflict of interest statement in the “Confidential to Editor” section, and submit your "Accept" recommendation.

Reviewer #3: (No Response)

Reviewer #4: (No Response)

Reviewer #5: (No Response)

2. Is the manuscript technically sound, and do the data support the conclusions?

Reviewer #3: Yes

Reviewer #4: Partly

Reviewer #5: Yes

3. Has the statistical analysis been performed appropriately and rigorously? 

Reviewer #3: Yes

Reviewer #4: Yes

Reviewer #5: Yes

4. Have the authors made all data underlying the findings in their manuscript fully available?

Reviewer #3: Yes

Reviewer #4: Yes

Reviewer #5: Yes

5. Is the manuscript presented in an intelligible fashion and written in standard English?

Reviewer #3: Yes

Reviewer #4: Yes

Reviewer #5: Yes

6. Review Comments to the Author

Reviewer #3: The statistical analysis and the methodology are well done. However, the safety assessment part (line 271-276) might need to be improved due to the outdated references; WHO (1991) and Markert (1992); and the lack of criteria, as only the permissible level of As, Cd and Pb were indicated.

Therefore, I recommend including additional permissible levels, such as from national regulations where the study was conducted or from regional regulations such as in Europe or the United States. This might help to consider the usage of this plant on an international scale.

Moreover, the consumption rate of medicinal plants is also an important factor. I suggest calculating some trace elements toxicity assessment indices that involved consumption rate such as Daily Intake of Metal (DIM) and Health Risk Index (HRI). These indices could help the readers to assess the safety of most metals. The consumption rate could be adapted from previous studies or by assumption to recommend the optimal consumption rate.

Reviewer #4: Dear Authors,

Its my pleasure to review your research and suggest comments aimed for its wider acceptability to researchers and scientists working in the field. I find the work novel and interesting. Few comments,

1. Abstract- Line number 30: 'reference plant' is a confusing term. What this implies? LIne number 30-31, need to be revised.

2. Abstract is vague. What are the investigated elements? Try to follow a model- 1-2 lines on background/need for the study, 2 lines on methodology, 3-4 lines on result and discussion (with some important data quotes from the study) and lastly 1 line on strong conclusion/recomendation. I will suggest the authors to reframe the abstract.

3. Introduction- My suggestion will be to bring the context from a 'global' to 'local' scale gradually, as readers may be from different geographical locations working on some other plant model but the conceptualisation of the research problem may be similar to the study presented. The most recent reference mentioned in the Introduction part is of Ziener et al., 2015; Hamza et al., 2015. This is not acceptable as per my view. Recent references need to be quoted. Globally there has been plethoras of work on pollution and its impact on ethnopharmacologically significant species- https://link.springer.com/article/10.1007/s11356-021-12566-w, https://link.springer.com/article/10.1007/s10653-020-00603-5, https://link.springer.com/article/10.1007/s10661-019-7714-7, https://www.tandfonline.com/doi/abs/10.1080/15226514.2018.1524841 etc.

Overall, I find the article novel and can be recomended for publication after authors addresses the comments given above.

Reviewer #5: The authors studied the trace element concentrations in medicinal plant Centaurium erythraea Rafn that growing on different polluted sites by using the instrumental neutron activation analysis (INAA). The manuscript is well-written. The writing, clarity and english language are good. Methods are accurate, Results and Discussion are related. In this form, I can recommend acceptance.

7. PLOS authors have the option to publish the peer review history of their article (what does this mean?). If published, this will include your full peer review and any attached files.

Reviewer #3: No

Reviewer #4: No

Reviewer #5: No

---

## [Author Response · Author response to Decision Letter 1]

21 Oct 2022

Reviewer #3:The statistical analysis and the methodology are well done. However, the safety assessment part (line 271-276) might need to be improved due to the outdated references; WHO (1991) and Markert (1992); and the lack of criteria, as only the permissible level of As, Cd and Pb were indicated. Therefore, I recommend including additional permissible levels, such as from national regulations where the study was conducted or from regional regulations such as in Europe or the United States. This might help to consider the usage of this plant on an international scale.

Moreover, the consumption rate of medicinal plants is also an important factor. 

***Thank you very much for submitting your comments.We have added appropriate corrections.

I suggest calculating some trace elements toxicity assessment indices that involved consumption rate such as Daily Intake of Metal (DIM) and Health Risk Index (HRI). These indices could help the readers to assess the safety of most metals. The consumption rate could be adapted from previous studies or by assumption to recommend the optimal consumption rate. 

***This is a medicinal plant, this plant material is not consumed every day, so we do not see the application of the above-mentioned factors.

Reviewer #4: Dear Authors,

Its my pleasure to review your research and suggest comments aimed for its wider acceptability to researchers and scientists working in the field. I find the work novel and interesting. Few comments,

1. Abstract- Line number 30: 'reference plant' is a confusing term. What this implies? LIne number 30-31, need to be revised.

2. Abstract is vague. What are the investigated elements? Try to follow a model- 1-2 lines on background/need for the study, 2 lines on methodology, 3-4 lines on result and discussion (with some important data quotes from the study) and lastly 1 line on strong conclusion/recomendation. I will suggest the authors to reframe the abstract.

3. Introduction- My suggestion will be to bring the context from a 'global' to 'local' scale gradually, as readers may be from different geographical locations working on some other plant model but the conceptualisation of the research problem may be similar to the study presented. The most recent reference mentioned in the Introduction part is of Ziener et al., 2015; Hamza et al., 2015. This is not acceptable as per my view. Recent references need to be quoted. Globally there has been plethoras of work on pollution and its impact on ethnopharmacologically significant species- https://link.springer.com/article/10.1007/s11356-021-12566-w, https://link.springer.com/article/10.1007/s10653-020-00603-5, https://link.springer.com/article/10.1007/s10661-019-7714-7, https://www.tandfonline.com/doi/abs/10.1080/15226514.2018.1524841 etc.

***Thank you very much for submitting your comments.We have added appropriate corrections.The text has been checked for language by a native speaker.

---

## [Editor Report · Decision Letter 2]

1 Nov 2022

PONE-D-21-34064R2Nuclear analytical techniques used to study the trace element content of Centaurium erythraea Rafn, a medicinal plant from sites with different pollution loads in Lower Silesia (SW Poland)PLOS ONE

Dear Dr. Kosior,

Thank you for submitting your manuscript to PLOS ONE. After careful consideration, we feel that it has merit but does not fully meet PLOS ONE’s publication criteria as it currently stands. Therefore, we invite you to submit a revised version of the manuscript that addresses the points raised during the review process.

The authors have never spent a word to respond to the Academic Editor's comments during two review rounds, neither they have fully addressed AE's comments. Since the manuscript still suffers from **numerous stylistic and grammar errors**, I urge the authors to have the manuscript professionally edited by a commercial English language editing service, since the manuscript will not be possibly accepted for publication in this form. **Some but not all** language flaws are listed below.

Please avoid vernacular phrases such as "a medicinal plant" (L2), "is a plant" (L21, L58), "This plant" (L59), "of this plant" (L70), "A daily intake of the medicinal plant" (L261; moreover, which plant?). Please use “a medicinal plant species” and similar.

L38: "The use of medical herbs to relieve and cure human diseases was in wide use in former times..." - please note that "the use" cannot be "in wide use".

L51-52: "...because of ongoing developments in nutrition and in biochemical surveying and mineral prospecting." - this part of the sentence is incomprehensible. Please rewrite to make sense.

L55: "toxicant element levels" - only "toxic elements' levels" or "levels of toxicants" can stand.

L60: Somewhere in this line has to be a full stop. Otherwise, the claim in L59-63 has no meaning.

L70: "...raw material of this plant are" - erroneous subject/verb agreement, a vernacular phrase.

L93: "...will allow a pattern and a relationship between the trace elements to be distinguished." - Please be clearer here since this is one of the main aims of the study.

L124 and further in the text: "15 min" instead of "15min".

L126 and further in the text: "4 days" instead of "4days".

L127-128: "...and measured twice, first at 4 days for 30 min and then at 20 days for 1.5hours." - I could not infer what the authors have meant to say here.

L130: "1332 keV" instead of "1332keV".

L131: "from Canberra" - please provide a reference or a link to the producer.

L192: Please write: "are given in Table 2." instead of "are given in (Table 2)."

L201 and further in the text: Please write: "given by Markert et al. [a reference number in a square bracket]" instead of "given by (Markert et al., 2015)".

L204: Please remove "the" from "for the *C. erythraea*"

L211-212: The sentence is too fuzzy to be understood. Please rewrite to clarify.

L240-242: Please avoid reprising "these elements" in the same sentence.

L250-251: "V will probably be part of the system protecting against injury to tissues" - please rewrite to clarify the meaning.

L259: Wrong conjunction. Please use a semicolon.

L276-279: Please rephrase the sentence to clarify its meaning.

L283: "the latter" - please precise what is "the latter".

L285-287: Please rewrite the whole sentence to clarify its meaning.

Moreover, some **text misinterpretations and methodological omissions** must be resolved in the revised manuscript. They are as follows:

L109: Were the samples cleaned of extraneous plant material only or the soil as well?

L133: I could not find this software in their site. Can you please provide a link? What the abbreviation "FLNP" represents?

L209-211: This text should stand in the table footnote, not here.

L214-215: Wasn't the same already said few sentences earlier?

L229: "Concentrations of REEs can exert..." - Please define which concentrations.

L238: The student's t test was not mentioned in the M&M section.

L245: Please note that Fe is the basic mineral contained in hemoglobin of our erythrocytes. Without it, we couldn't exchange oxygen and CO_2_ with our environment.

L247-248: I cannot resolve the meaning of this sentence. Red blood cells are necessary for rapid oxygen transport to injured cells? What is produced in the body?

L261: "A daily intake of the medicinal plant..." - please define which medicinal plant.

L263: "plus others" - please avoid using vernacular expressions in scientific literature. What others? Please define.

L284: "has to be verified" - Where it should be verified? In this study?

L303: Reference list and citations in the text should be formatted according to the Journal submission guidelines.

L490: "Levene's test is significant" - please note that this test was not mentioned in the M&M section.

**Additional efforts** should be put to argumentatively respond to the Reviewer#1's suggestion:

"I suggest calculating some trace elements toxicity assessment indices that involved consumption rate such as Daily Intake of Metal (DIM) and Health Risk Index (HRI). These indices could help the readers to assess the safety of most metals. The consumption rate could be adapted from previous studies or by assumption to recommend the optimal consumption rate."

The authors argue that "This is a medicinal plant, this plant material is not consumed every day, so we do not see the application of the above-mentioned factors."

However, in L86-88, they state "The results from this study allow the dosage for medicines prepared from *C. erythraea* to be specified and the degree of risk of elements considered harmful to consumers." This is contradictory to their answer to the reviewer's concern.

The whole section "Statistical analysis", starting from L143, should be considerably condensed. It contains text repetitions and off-the-scope sentences. Please rewrite.

We look forward to receiving your revised manuscript.

Kind regards,

Branislav T. Šiler, Ph.D.

Academic Editor

PLOS ONE
---

## [Author Response · Author response to Decision Letter 2]

11 Feb 2023

Thank you very much for submitting your comments. We agree with all of the suggestions and have made major changes to the text to meet your requests.

---

## [Editor Report · Decision Letter 3]

15 Feb 2023

PONE-D-21-34064R3Nuclear analytical techniques used to study the trace element content of Centaurium erythraea Rafn, a medicinal plant species from sites with different pollution loads in Lower Silesia (SW Poland)PLOS ONE

Dear Dr. Kosior,

Thank you for submitting your manuscript to PLOS ONE. After careful consideration, we feel that it has merit but does not fully meet PLOS ONE’s publication criteria as it currently stands. Therefore, we invite you to submit a revised version of the manuscript that addresses the points raised during the review process.

The Academic Editor's recommendation from the previous review round: "The whole section "Statistical analysis", starting from L143, should be considerably condensed. It contains text repetitions and off-the-scope sentences. Please rewrite." remained unaddressed. The whole section is overloaded with repetitive and off-topic text and it should be thoroughly reorganized.

The second major problem with the text is that it is still abundant with stylistic and grammar inconsistencies. Here are some examples:

L40-41: "The use of medical herbs to relieve and cure human diseases was in use in former times, but their applications decreased with introduction of modern medicine." - The sentence refers to the "use of use" in its first part. Please rewrite it to clarify.

L61-63: "*C. erythraea* is widespread in Europe and parts of western Asia and northern Africa, where the aerial parts are collected *C. erythraea* occurs mostly in open habitats with bare areas." - The sentence is grammatically incorrect.

L266-268: "The total level of these elements in the plant permits us to conclude that *C. erythraea* can be considered as an important their source for humans" - The sentence is vague.

L290-291: "For humans, elements e.g. Zn, Fe, Cu, Cr, Mo, Se and Co, are considered as essential" - the sentence is grammatically incorrect.

L311-312: "These undesirable elements, potentially exhibiting toxic effects." - The sentence is grammatically incorrect.

For the fourth time, I urge the authors to have the manuscript professionally edited by a native English speaker or a professional editing agency. Please note that "PLOS ONE does not copyedit accepted manuscripts, so the language in submitted articles must be clear, correct, and unambiguous. We may reject papers that do not meet these standards." and "Submissions that do not meet the PLOS ONE publication criterion for language standards may be rejected.", according to PLOS ONE's Criteria for Publication (https://journals.plos.org/plosone/s/criteria-for-publication#loc-5).

We look forward to receiving your revised manuscript.

Kind regards,

Branislav T. Šiler, Ph.D.

Academic Editor

PLOS ONE
---

## [Author Response · Author response to Decision Letter 3]

15 Apr 2023

Thank you very much for submitting your comments. We agree with all of the reviewers' suggestions and have made major changes to the text to meet your requests.

---

## [Editor Report · Decision Letter 4]

20 Apr 2023

Nuclear analytical techniques used to study the trace element content of Centaurium erythraea Rafn, a medicinal plant species from sites with different pollution loads in Lower Silesia (SW Poland)

PONE-D-21-34064R4

Dear Dr. Kosior,

We’re pleased to inform you that your manuscript has been judged scientifically suitable for publication and will be formally accepted for publication once it meets all outstanding technical requirements.

Kind regards,

Branislav T. Šiler, Ph.D.

Academic Editor

PLOS ONE
---

## [Editor Report · Acceptance letter]

9 May 2023

PONE-D-21-34064R4 

Nuclear analytical techniques used to study the trace element content of *Centaurium erythraea* Rafn, a medicinal plant species from sites with different pollution loads in Lower Silesia (SW Poland) 

Dear Dr. Kosior:

I'm pleased to inform you that your manuscript has been deemed suitable for publication in PLOS ONE. Congratulations! Your manuscript is now with our production department. 

Kind regards, 

on behalf of

Dr. Branislav T. Šiler 

Academic Editor

PLOS ONE